# An Adaptive Mirror-Prox Algorithm for Variational Inequalities with Singular Operators

**Kimon Antonakopoulos**
Univ. Grenoble Alpes, CNRS, Inria, Grenoble INP
LIG 38000 Grenoble, France.
kimon.antonakopoulos@inria.fr

**E. Veronica Belmega**
ETIS/ENSEA
Univ. de Cergy-Pontoise-CNRS, France
belmega@ensea.fr

**Panayotis Mertikopoulos**
Univ. Grenoble Alpes, CNRS, Inria, Grenoble INP,
LIG 38000 Grenoble, France.
panayotis.mertikopoulos@imag.fr

## Abstract

Lipschitz continuity is a central requirement for achieving the optimal $\mathcal{O}(1/T)$ rate of convergence in monotone, deterministic variational inequalities (a setting that includes convex minimization, convex-concave optimization, nonatomic games, and many other problems). However, in many cases of practical interest, the operator defining the variational inequality may exhibit singularities at the boundary of the feasible region, precluding in this way the use of fast gradient methods that attain this optimal rate (such as Nemirovski's mirror-prox algorithm and its variants). To address this issue, we consider a regularity condition which relates the variation of the operator to that of a suitably chosen Bregman function. Leveraging this *Bregman continuity* condition, we derive an adaptive mirror-prox algorithm which attains the optimal $\mathcal{O}(1/T)$ rate of convergence in problems with possibly singular operators, without any prior knowledge of the degree of smoothness (the Bregman analogue of the Lipschitz constant). We also show that, under Bregman continuity, the mirror-prox algorithm achieves a $\mathcal{O}(1/\sqrt{T})$ convergence rate in stochastic variational inequalities.

## 1 Introduction

The seminal introduction of generative adversarial networks (GANs) [18] has ushered in a new optimization paradigm in deep learning: instead of focusing on the minimization of an empirical loss function, GAN training hinges on a zero-sum game between a generator and a discriminator. In fact, in many cases GAN training goes even beyond the min-max setting, either because there are more than two networks involved, or because the objectives of the generator and the discriminator are not entirely opposed – e.g., as in the widely used ACGAN framework of Odena et al. [43]. In these cases, the most compact way of representing the problem's training landscape is by means of a *variational inequality* (VI).

Tracing their origins to the work of Stampacchia [49] on the Signorini problem, variational inequalities have since found a broad range of applications in physics, engineering, economics – and, more recently, machine learning. One of the main reasons for their extensive applicability is that they comprise a flexible optimization framework which can simultaneously account for loss function minimization, saddle-point, game-theoretic, and fixed point problems. As a result, there has been considerable interest in the literature to develop optimal algorithms for solving VI problems; for an appetizing introduction, see [16] and references therein.

One of the most widely studied methods for this purpose is ordinary gradient descent – also known as the forward-backward (FB) algorithm in operator theory [6].[1] In monotone, deterministic variational inequalities, the convergence of the method is guaranteed under a condition known as *cocoercivity*. By the Baillon–Haddad theorem, if the operator defining the variational inequality is a gradient field (i.e., in loss minimization problems), this condition is equivalent to Lipschitz smoothness of the associated loss function [4, 6]. However, cocoercivity may fail to hold even in simple, bilinear min-max problems, in which case gradient descent provably fails to converge – see e.g., [17, 35, 36] for a precise statement.

The first algorithm achieving convergence in (pseudo-)monotone variational inequalities without cocoercivity is the *extra-gradient* (EG) algorithm of Korpelevich [24], which only requires Lipschitz continuity of the underlying operator.[2] The asymptotic convergence result of Korpelevich [24] was subsequently extended by Nemirovski [38] who introduced the *mirror-prox* (MP) algorithm, a Bregman variant of the EG algorithm with ergodic averaging. As was shown in [38], the mirror-prox algorithm attains a $\mathcal{O}(1/T)$ ergodic convergence rate in monotone variational inequalities with Lipschitz continuous operators, and this rate cannot be improved without further assumptions.

However, in many applications and problems of practical interest, Lipschitz continuity may also fail to hold, either because the loss profile of the problem grows too rapidly (e.g., as in support vector machines or GAN models with Kullback-Leibler losses), or because the problem exhibits singularities near the boundary of the feasible region (e.g., as in resource allocation and inverse problems). In these cases, one would still want to apply a fast method like mirror-prox, but the lack of smoothness means that there are no convergence guarantees – asymptotically, ergodically, or otherwise.

**Our contributions.** Our starting point is the observation that this failure stems from the fact that Lipschitz continuity of the operator is defined relative to a *global* norm. Because of this, the standard Lipschitz framework is not well-suited to problems with singularities or rapid growth: a global norm is oblivious to the geometry of the feasible region (and, in particular, its boundary), so it cannot capture the finer features of the problem's loss landscape.

To overcome this limitation, we introduce a novel regularity condition, which we call *Bregman continuity*, and which is made-to-order for the singularity landscape of the problem at hand. Specifically, instead of defining Lipschitz continuity relative to a global norm, we define it in terms of a family of *local* norms and a suitably chosen *Bregman function*. This leads to an intricate interplay between different geometric notions of distance (the Bregman divergence and the local norm), but it also introduces the flexibility required to tackle variational inequalities with singular operators.

Under this assumption, we show that the mirror-prox algorithm attains the optimal $\mathcal{O}(1/T)$ convergence rate in variational inequalities with (possibly) singular operators, provided that the method is run with the same Bregman function that is used to define Bregman continuity. As in the standard Lipschitz framework, the method's convergence requires a step-size of the form $\gamma < 1/\beta$, where $\beta$ is the Bregman constant of the operator (i.e., the Bregman analogue of the Lipschitz constant). Estimating this constant can be fairly challenging in practice (if not downright impossible), so we also introduce an *adaptive mirror-prox* (AMP) method which attains the same $\mathcal{O}(1/T)$ rate without requiring any a priori estimation of $\beta$ – essentially, the Bregman constant is learned at the same time as the problem's landscape. Finally, we provide a variant of the method for stochastic variational inequalities, and we establish a $\mathcal{O}(1/\sqrt{T})$ convergence rate in this setting. To the best of our knowledge, these are the first results of this kind in the literature.

**Related work.** Owing to their optimal rate guarantees, the extra-gradient and mirror-prox algorithms have been at the forefront of an extensive literature which is impossible to adequately review here. As a purely indicative list of contributions in the Lipschitz continuous setting (and with no illusion of being comprehensive), we refer the reader to Juditsky et al. [21], Chambolle and Pock [13], Malitsky [28], Iusem et al. [20] and Mokhtari et al. [37] for some recent developments. Especially in

learning theory, there has been a surge of interest motivated by the application of EG/MP methods to GAN training, see e.g., [15, 17, 36, 51] and references therein.[3]

Going beyond the Lipschitz regime, Bauschke et al. [7] recently introduced a "Lipschitz-like" smoothness condition for convex minimization problems and used it to establish a $\mathcal{O}(1/T)$ value convergence rate for mirror descent methods (as opposed to mirror-prox). Always in the context of loss minimization problems, Bolte et al. [9] subsequently extended the results of Bauschke et al. [7] to unconstrained non-convex problems that satisfy the Kurdyka–Łojasiewicz (KL) inequality, while Lu et al. [27] considered functions that are also relatively strongly convex and showed that mirror descent achieves a geometric convergence rate in this context. Finally, in a very recent preprint, Hanzely et al. [19] examined the rate of convergence of an accelerated variant of mirror descent under the same Lipschitz-like smoothness assumption.

The condition of Bauschke et al. [7] is remarkably simple as it only posits that the problem's loss function $f$ is such that $\beta h - f$ is convex for some reference Bregman function $h$ and some $\beta > 0$. A straightforward extension of this condition to an operator setting would be to require the monotonicity of $\beta \nabla h - A$, where $A$ is the operator defining the variational inequality under study. However, the cornerstone of this "Lipschitz-like" condition is a descent lemma which does not carry over to variational inequalities, so it does not seem possible to extend the analysis of Bauschke et al. [7] to an operator setting. Lu [26] also considered a "relative continuity" condition for loss minimization problems positing that $\|\nabla f(x)\| \leq M \inf_{x'} \sqrt{2D(x', x)} / \|x' - x\|$ (where $f$ is the problem's objective and $D$ is the Bregman divergence of $h$). Written this way, the condition of Lu [26] can also be extended to an operator setting, but this would provide a surrogate for operator *boundedness*, not Lipschitz continuity (since $A = \nabla f$ in minimization problems). Since the optimal $\mathcal{O}(1/T)$ convergence rate of the mirror-prox algorithm is tied to the *regularity* of $A$ – as opposed to its boundedness – the condition of Lu [26] does not seem applicable to the setting under study. Accordingly, there is no overlap in results or methodology with this particular strand of the literature.

Finally, in a very recent paper, Bach and Levy [3] introduced a *universal* variant of the mirror-prox algorithm which is model-agnostic and achieves an optimal convergence rate in stochastic and/or smooth settings. Achieving optimal rates in the setting of Bach and Levy [3] relies crucially on the operator being Lipschitz continuous (albeit with a possibly unknown constant) and the feasible region having a finite Bregman diameter. The algorithm we propose in this work is not universal but it *is* adaptive, and it does not require either Lipschitz continuity or a finite Bregman diameter. In this manner, our work also provides an important first step towards extending the universal analysis of Bach and Levy [3] to VI problems with singularities.

## 2  Preliminaries

Let $\mathcal{X}$ be a convex – but not necessarily closed or compact – subset of a $d$-dimensional normed space $\mathcal{V}$, and let $\mathcal{V}^*$ denote the dual space of $\mathcal{V}$. The *variational inequality* (VI) problem associated to a continuous operator $A \colon \mathcal{X} \to \mathcal{V}^*$ consists of finding $x^* \in \mathcal{X}$ such that

$$\langle A(x^*), x - x^* \rangle \geq 0 \text{ for all } x \in \mathcal{X}. \tag{VI}$$

Following [16], we will refer to this problem as $\mathrm{VI}(\mathcal{X}, A)$ and we will write $\mathcal{X}^* \equiv \mathrm{Sol}(\mathcal{X}, A)$ for its set of solutions. Note also that, if $\mathcal{X}$ is not closed, $A$ may exhibit a *singularity* at a residual point $x \in \mathrm{bd}(\mathcal{X}) \setminus \mathcal{X}$ in the sense that $A$ does not admit a continuous extension to $x$.

In the literature, this formulation of the problem is often referred to as a *Stampacchia variational inequality* (SVI) [16] or a "strong" variational inequality [21, 40]. For illustration purposes, we present some archetypal examples of such problems below:

**Example 2.1** (Loss minimization). If $A = \nabla f$ for some convex loss function $f$ on $\mathcal{X} = \mathbb{R}^d$, solutions of (VI) coincide with the global minimizers of $f$.

**Example 2.2** (Min-max optimization). Suppose that $A = (\nabla_{x_1} f, -\nabla_{x_2} f)$ for some real-valued function $f(x_1, x_2)$ with $x_1 \in \mathcal{X}_1, x_2 \in \mathcal{X}_2$. If $f$ is convex-concave (i.e., convex in $x_1$ and concave in

$x_2$), any solution $x^* = (x_1^*, x_2^*)$ of (VI) is a global saddle-point of $f$, i.e.,
$$f(x_1^*, x_2^*) \le f(x_1, x_2^*) \quad \text{and} \quad f(x_1^*, x_2^*) \ge f(x_1^*, x_2) \tag{2.1}$$
for all $x_1 \in \mathcal{X}_1, x_2 \in \mathcal{X}_2$. Problems of this type have attracted considerable interest in the fields of machine learning and artificial intelligence because they constitute the basic optimization framework for GANs [18]. For a series of recent papers focusing on the interplay between GAN and saddle-point problems / variational inequalities, see [15, 17, 25, 36, 51] and references therein.

**Example 2.3** (Resource sharing problems). Consider a set of *resources* $r \in \mathcal{R} = \{1, \ldots, R\}$ serving a stream of *demands* that arrive at a rate of $\rho$ per unit of time (for instance, a GPU cluster or a computing grid processing a stream of jobs). If the load on the $r$-th resource is $x_r$, the expected service time in the standard Kleinrock model [23] is given by the M/M/1 loss function
$$\ell_r(x_r) = \frac{1}{c_r - x_r}, \tag{2.2}$$
where $c_r$ denotes the capacity of the resource. In this setting, the set of feasible resource allocations is $\mathcal{X} \equiv \{(x_1, \ldots, x_R) : 0 \le x_r < c_r, x_1 + \cdots + x_R = \rho\}$,[4] and we say that a resource allocation profile $x^* \in \mathcal{X}^*$ is at *Nash/Wardrop equilibrium* [42, 50] if
$$\ell_r(x_r^*) \le \ell_r(x_r) \quad \text{for all } x \in \mathcal{X} \text{ and all } r \in \mathcal{R} \text{ such that } x_r^* > 0 \tag{2.3}$$
i.e., when no job would be better served by transferring it to a different priority queue. In this case, if we let $A(x) = (\ell_1(x_1), \ldots, \ell_R(x_R))$, a standard calculation shows that $x^*$ is an equilibrium allocation if and only if it solves the associated variational inequality problem for $A$.

The most widely used assumption in the literature for solving VI problems is *monotonicity*, i.e.,
$$\langle A(x') - A(x), x' - x \rangle \ge 0 \quad \text{for all } x, x' \in \mathcal{X}. \tag{2.4}$$
When $A = \nabla f$, this condition is equivalent to $f$ being convex; likewise, when $A = (\nabla_{x_1} f, -\nabla_{x_2} f)$ as in Example 2.2, monotonicity is equivalent to $f$ being convex-concave [6]; finally, by direct calculation, it is straightforward to see that the operator defined in Example 2.3 is monotone. For an introduction to the theory of monotone operators, we refer the reader to Facchinei and Pang [16] and Bauschke and Combettes [6].

Now, drawing on Nesterov [40, 41] and Juditsky et al. [21], if $A$ is monotone, the quality of a candidate solution $\hat{x} \in \mathcal{X}$ can be assessed via the *restricted gap* (or *merit*) *function*
$$\mathrm{Gap}_{\mathcal{C}}(\hat{x}) = \sup_{x \in \mathcal{C}} \langle A(x), \hat{x} - x \rangle, \tag{2.5a}$$
where $\mathcal{C}$ is a nonempty convex subset of $\mathcal{X}$. The rationale behind this definition is that, if $x^*$ solves (VI), monotonicity gives $\langle A(x), x^* - x \rangle \le \langle A(x^*), x^* - x \rangle \le 0$, so the quantity being maximized in (2.5) is small if $\hat{x}$ is an approximate solution of (VI). Formally, we have:

**Lemma 1.** *Suppose that $A$ is monotone. If $x^*$ solves* (VI)*, we have* $\mathrm{Gap}_{\mathcal{C}}(x^*) = 0$ *whenever* $x^* \in \mathcal{C}$. *Conversely, if* $\mathrm{Gap}_{\mathcal{C}}(\hat{x}) = 0$ *and $\mathcal{C}$ contains a neighborhood of $\hat{x}$ in $\mathcal{X}$, $\hat{x}$ is a solution of* (VI)*.*

This lemma extends a similar result by Nesterov [40], so we defer its proof to the paper's supplement. In view of all this, we will employ the gap function $\mathrm{Gap}_{\mathcal{C}}(\hat{x})$ as our main figure of merit and we will use it to state our convergence rate guarantees in the sequel.

## 3 Bregman continuity

In addition to monotonicity, a standard assumption for solving variational inequalities is that of *Lipschitz continuity*, i.e.,
$$\|A(x') - A(x)\|_* \le \beta \|x' - x\| \tag{Lip}$$
for some $\beta > 0$ and for all $x, x' \in \mathcal{X}$. This definition involves two distinct (but related) measures of distance: (*i*) the primal norm on $\mathcal{V}$ which measures distances between the primal points $x, x' \in \mathcal{X}$; and (*ii*) the dual norm on $\mathcal{V}^*$ which measures the distance between the dual vectors $A(x), A(x') \in \mathcal{V}^*$.[5]

Importantly, both of these notions are *global*, i.e., they do not depend on the point in space at which they are calculated; as such, Lipschitz continuity is oblivious to the geometry of $\mathcal{X}$ (and, in particular, its boundary). In the sequel, we describe a way to overcome this limitation by introducing two distinct notions of distance that are tailored to the geometry of $\mathcal{X}$ and the singularity landscape of $A$.

**Local norms.** The first measure of distance that we define is that of *local norm* on $\mathcal{X}$:

**Definition 1.** Let $\mathcal{Z} = \mathrm{span}(\mathcal{X} - \mathcal{X})$ denote the *tangent hull* of $\mathcal{X}$, i.e., the subspace of $\mathcal{V}$ spanned by all possible displacement vectors of the form $z = x' - x$, $x, x' \in \mathcal{X}$. A *local norm* on $\mathcal{X}$ is a continuous assignment of a norm $\|\cdot\|_x$ on $\mathcal{Z}$ at each $x \in \mathcal{X}$.[6] The induced *dual local norm* is then defined as

$$\|v\|_{x,*} = \max_{z \in \mathcal{Z}}\{|\langle v, z\rangle| : \|z\|_x \leq 1\} \quad \text{for all } v \in \mathcal{V}^*. \tag{3.1}$$

For ease of presentation, we tacitly assume in what follows that $\|z\|_x \geq \mu\|z\|$ for some $\mu > 0$ and all $x \in \mathcal{X}$, $z \in \mathcal{Z}$. This can always be achieved by taking $\|\cdot\|_x \leftarrow \|\cdot\|_x + \mu\|\cdot\|$ so there is no loss of generality. Note in particular that this implies that $\|v\|_{x,*} \leq (1/\mu)\|v\|$ for all $x \in \mathcal{X}$ and all $v \in \mathcal{Z}^*$.

For intuition, we present some key examples below:

**Example 3.1** (Euclidean geometry). Let $\mathcal{X} = \mathbb{R}^d$ so $\mathcal{Z} = \mathbb{R}^d$. The *Euclidean norm* on $\mathcal{X}$ is given by the standard expression $\|z\|_2^2 = \sum_{j=1}^d z_j^2$, and the associated dual norm is the same.

**Example 3.2** (Shahshahani $p$-norm). Let $\mathcal{X} = \mathbb{R}^d_{++}$ so, again, $\mathcal{Z} = \mathbb{R}^d$. The *Shahshahani $p$-norm* on $\mathcal{X}$ is defined for all $p > 1$ as

$$\|z\|_x = \left(|z_1|^p/x_1 + \cdots + |z_d|^p/x_d\right)^{1/p} \quad \text{for all } x \in \mathcal{X}, z \in \mathcal{Z}. \tag{3.2}$$

By a straightforward application of Hölder's inequality, the corresponding dual norm is given by

$$\|v\|_{x,*} = \left(x_1^{q-1}|v_1|^q + \cdots + x_d^{q-1}|v_d|^q\right)^{1/q} \tag{3.3}$$

with the usual convention $p^{-1} + q^{-1} = 1$. In particular, for $p \to 1^+$, we get the limiting expression

$$\|v\|_{x,*} = \max\{x_1|v_1|, \ldots, x_d|v_d|\}. \tag{3.4}$$

This metric plays a major role in, among others, game theory, optimal transport, machine learning, information theory, and many other fields – see e.g., [1, 2, 22, 31, 34, 47, 48] and references therein.

**Local Bregman functions and the associated divergence.** The notion of a dual local norm presented above will be our principal measure of distance in $\mathcal{V}^*$. To proceed, we will also need to adapt the notion of a *Bregman* (or *distance-generating*) function on $\mathcal{X}$:

**Definition 2.** Let $\|\cdot\|_x$ be a local norm on $\mathcal{X}$. We say that $h\colon \mathcal{V} \to \mathbb{R}$ is a *Bregman function* on $\mathcal{X}$ if:

1. $h$ is proper, l.s.c., convex, and $\mathrm{dom}\, h = \mathcal{X}$.

2. The subdifferential of $h$ admits a *continuous selection*, i.e., a continuous function $\nabla h$ such that $\nabla h(x) \in \partial h(x)$ for all $x \in \mathcal{X}^\circ \equiv \mathrm{dom}\, \partial h$.

3. $h$ is strongly convex relative to the underlying local norm, i.e.,

$$h(p) \geq h(x) + \langle \nabla h(x), p - x\rangle + \tfrac{1}{2}K\|p - x\|_x^2 \tag{3.5}$$

   for some $K > 0$ and all $p \in \mathcal{X}$, $x \in \mathcal{X}^\circ$.

The *Bregman divergence* induced by $h$ is then defined for all $p \in \mathcal{X}$, $x \in \mathcal{X}^\circ$, as

$$D(p, x) = h(p) - h(x) - \langle \nabla h(x), p - x\rangle. \tag{3.6}$$

As an immediate consequence of the above, we have:

**Lemma 2.** *A Bregman function $h$ is $K$-strongly convex relative to $\|\cdot\|_x$ if and only if*

$$D(p, x) \geq \tfrac{1}{2}K\|p - x\|_x^2 \quad \text{for all } p \in \mathcal{X} \text{ and all } x \in \mathcal{X}^\circ. \tag{3.7}$$

The main difference between Definition 2 and the standard assumptions in the literature [7, 10, 11, 21, 30, 32, 33, 39–41] is the strong convexity requirement relative to the local norm $\|\cdot\|_x$ (whose choice, in turn, is aimed to capture the singularity landscape of the operator). We illustrate this with two examples below:

**Example 3.3.** Suppose that $\mathcal{X} = \mathbb{R}^d$ is endowed with the Euclidean norm as in Example 3.1. Then, setting $h(x) = (1/2)\|x\|_2^2$, we get the standard expression $D(p, x) = (1/2)\|p - x\|_2^2$ for the associated Bregman divergence. Obviously, $h$ is 1-strongly convex relative to $\|\cdot\|_2$.

**Example 3.4.** Let $\mathcal{X} = [0, 1)^d$ (so $\mathcal{X}$ is neither open nor closed), and consider the local norm $\|z\|_x^2 = \sum_{i=1}^d |z|_i^2/(1 - x_i)^2$ for $x \in \mathcal{X}$, $z \in \mathbb{R}^d$ (cf. Example 3.2 above). If we set

$$h(x) = \sum_{i=1}^d 1/(1 - x_i) \tag{3.8}$$

a straightforward calculation gives

$$D(p, x) = \sum_{i=1}^d \frac{(p_i - x_i)^2}{(1 - p_i)(1 - x_i)^2} \geq \sum_{i=1}^d \frac{(p_i - x_i)^2}{(1 - x_i)^2} = \|p - x\|_x^2, \tag{3.9}$$

i.e., $h$ is strongly convex relative to $\|\cdot\|_x$. Importantly, since $\|\cdot\|_x \geq \|\cdot\|_2$, this Bregman function is also strongly convex relative to the standard Euclidean norm. However, even though the Euclidean regularizer of Example 3.3 is strongly convex relative to *any* global norm on $\mathcal{X}$, it cannot be strongly convex relative to the local norm $\|\cdot\|_x$ because of the singularity of the latter when $x_i \to 1^-$.

**Bregman continuity.** We are now in a position to introduce the notion of *Bregman continuity:*

**Definition 3.** Let $h$ be a local Bregman function relative to some local norm $\|\cdot\|_x$ on $\mathcal{X}$. We say that the operator $A : \mathcal{X} \to \mathcal{V}^*$ is $\beta$-*Bregman continuous* if

$$\|A(x') - A(x)\|_{x,*} \leq \beta\sqrt{2D(x, x')} \quad \text{for all } x, x' \in \mathcal{X}. \tag{BC}$$

Of course, in the case of a global norm with Bregman function $h(x) = (1/2)\|x\|^2$ (cf. Example 3.3), we recover the standard Lipschitz continuity condition: $\|A(x') - A(x)\|_* \leq \beta\sqrt{2D(x', x)} = \beta\|x' - x\|$. On the other hand, the example below shows that an operator can be Bregman continuous without being Lipschitz continuous relative to *any* global norm:

**Example 3.5.** Consider the operator $A(x) = (c_r/(1 - x_r/c_r))_{r \in \mathcal{R}}$ defined in Example 2.3. Renormalizing $c_r$ to 1 for clarity and using the Bregman data of Examples 3.2 and 3.4, we get:

$$\|A(x') - A(x)\|_{x,*}^2 = \sum_{i=1}^d \frac{(x_i' - x_i)^2}{(1 - x_i')^2} \leq \sum_{i=1}^d \frac{(x_i' - x_i)^2}{(1 - x_i)(1 - x_i')^2} = D(x, x') \tag{3.10}$$

i.e., $A$ is $(1/\sqrt{2})$-Bregman continuous relative to $h$. However, given the singularity of $A(x)$ as $x_i \to 1^-$, we see that $A$ cannot be Lipschitz continuous relative to *any* global norm on $\mathcal{X}$.

Importantly, this example suggests the following rule of thumb: if the Jacobian of $A$ exhibits a singularity of the form $\mathcal{O}(\phi(x))$ near the residual set $\mathrm{cl}(\mathcal{X}) \setminus \mathcal{X}$ of $\mathcal{X}$, taking $\|\cdot\|_x = \Theta(\phi(x))$ and $h(x) = \Theta(\phi(x))$ allows $A$ to be Bregman continuous, despite this singularity. This heuristic provides a principled choice of Bregman data under which $A$ satisfies (BC).

## 4 The mirror-prox algorithm

In this section, we present the main algorithmic method that we will use to solve (VI) under Bregman continuity. Our core assumptions in that regard will be:

**Assumption 1.** The solution set $\mathcal{X}^* \equiv \mathrm{Sol}(\mathcal{X}, A)$ of (VI) is nonempty.

**Assumption 2.** $A$ is monotone and $\beta$-Bregman continuous.

In addition to the above, we assume that the optimizer gains access to $A$ via an *oracle* which, when called at the $t$-th stage of a sequence $X_t \in \mathcal{X}$, returns (possibly imperfect) feedback of the form

$$V_t = A(X_t) + U_t, \tag{4.1}$$

where $U_t \in \mathcal{V}^*$ is an additive noise variable. The two cases of interest that we consider here are (*i*) when $U_t = 0$ for all $t$; and (*ii*) when $U_t$ satisfies the statistical hypotheses:

    *a) Zero-mean:*        $\mathbb{E}[U_t \mid \mathcal{F}_t] = 0.$                                               (4.2a)

*b) Finite variance:*   $\mathbb{E}[\|U_t\|_*^2 \mid \mathcal{F}_t] \leq \sigma^2.$ (4.2b)

with $\mathcal{F}_t$ denoting the history (natural filtration) of $X_t$. For obvious reasons, we will refer to the first case ($U_t = 0$) as a *perfect oracle*, and to the second one as a *stochastic oracle*.

Following Nemirovski [38] and Juditsky et al. [21], the *mirror-prox* (MP) algorithm can be stated in recursive form as follows:

$$\begin{aligned} X_{t+1/2} &= P_{X_t}(-\gamma_t V_t) \\ X_{t+1} &= P_{X_t}(-\gamma_t V_{t+1/2}) \end{aligned} \tag{MP}$$

where $\gamma_t > 0$ is a variable step-size sequence (discussed in detail below), and the so-called "prox-mapping" $P\colon \mathcal{X}^\circ \times \mathcal{V}^* \to \mathcal{X}$ is defined as

$$P_x(y) = \arg\min_{x' \in \mathcal{X}}\{\langle y, x - x'\rangle + D(x', x)\} \tag{4.3}$$

with $D(\cdot, \cdot)$ denoting the divergence of an underlying Bregman function $h\colon \mathcal{X} \to \mathbb{R}$. For concreteness, we also assume in what follows that (MP) is initialized at the so-called "prox-center" of $\mathcal{X}$, i.e.,

$$X_1 = x_c \equiv \arg\min_{x \in \mathcal{X}} h(\mathcal{X}). \tag{4.4}$$

*Remark* 1. In general, calculating mirror steps can be computationally expensive – just like Euclidean projections in several cases. In what follows, we tacitly assume that our setting is "prox-friendly" [21, 38, 40] in the sense that the update (4.3) can be computed efficiently (e.g., as in Example 3.4).

Heuristically, the main idea behind (MP) is that, at each $t = 1, 2, \dots$, the oracle is called at the algorithm's base state $X_t$ to generate an intermediate, *leading* state $X_{t+1/2}$; subsequently, the base state is updated with oracle information from the leading state $X_{t+1/2}$ and the process repeats. In this way, (MP) essentially tries to "anticipate" the change of $A$ along a prox-step, and to exploit this "forward" information in order to achieve a faster convergence rate than ordinary forward-backward/gradient descent schemes. For this anticipatory scheme to work, the variation of the operator $A$ must be sufficiently gradual, hence the need for Lipschitz continuity in the classical analysis of the algorithm [21, 38, 40]. If this variation is unbounded (e.g., if $A$ exhibits singularities), this look-ahead mechanism could break down completely and the algorithm might fail to converge altogether. Our first result below is that, despite such singularities, Bregman continuity allows us to recover the optimal convergence rate of (MP):

**Theorem 1.** *Assume that $A$ satisfies Assumptions 1 and 2, and let $\mathrm{Gap}_H$ denote the restricted gap function for the Bregman zone $\mathcal{C}_H = \{x \in \mathcal{X} : D(x, x_c) \leq H\}$. Suppose further that (MP) is run with a $K$-strongly convex Bregman function and oracle feedback of the form (4.1). Then, for all $H > 0$, the averaged sequence $\bar{X}_T = \sum_{t=1}^T \gamma_t X_{t+1/2}\big/ \sum_{t=1}^T \gamma_t$ enjoys the following gap bounds:*

*a) If $\sigma^2 = 0$ and the algorithm's step-size satisfies*

$$0 < \gamma_{\min} \equiv \inf_t \gamma_t \leq \sup_t \gamma_t \equiv \gamma_{\max} \leq \sqrt{K}/\beta, \tag{4.5}$$

*we have*

$$\mathrm{Gap}_H(\bar{X}_T) \leq \frac{H}{\gamma_{\min}} \frac{1}{T} \tag{4.6}$$

*b) Otherwise, if $\sigma^2 > 0$ and $\gamma_t \leq \sqrt{K/2}/\beta$, we have*

$$\mathbb{E}[\mathrm{Gap}_H(\bar{X}_T)] = \mathcal{O}\left(\frac{H + \sigma^2 \sum_{t=1}^T \gamma_t^2}{\sum_{t=1}^T \gamma_t}\right) \tag{4.7}$$

*In particular, if $\gamma_t \propto 1/\sqrt{T}$, we get $\mathbb{E}[\mathrm{Gap}_H(\bar{X}_t)] = \mathcal{O}(1/\sqrt{T})$.*

As we show in the supplement, the key step in the proof of the deterministic part of Theorem 1 is the following energy inequality for an arbitrary target point $p \in \mathcal{C}_H$:

$$D(p, X_{t+1}) \leq D(p, X_t) - \gamma_t \langle A(X_{t+1/2}), X_{t+1/2} - p\rangle - \left(1 - \frac{\beta^2 \gamma_t^2}{K}\right) D(X_{t+1/2}, X_t) \tag{4.8}$$

There are two points where the Bregman structure of the algorithm can be seen in (4.8): in the energy iterates $D(p, X_t)$, but also in the comparison of the algorithm's base and leading state in the term $D(X_{t+1/2}, X_t)$. In the "vanilla" setting, Lipschitz continuity is used to obtain a comparison of these

---

**Algorithm 1:** adaptive mirror-prox (AMP)

---

**Require:** local norm $\|\cdot\|_x$, $K$-strongly convex Bregman function $h$, shrink ratio $\theta \in (0, 1)$

1: take $X_1 = \arg\min h$, $\gamma_1 > 0$             # initialization
2: **for** $t = 1, 2, \ldots$ **do**
3:     get oracle feedback $V_t$ at $X_t$           # base state query
4:     set $X_{t+1/2} = P_{X_t}(-\gamma_t V_t)$         # leading state update
5:     get oracle feedback $V_{t+1/2}$ at $X_{t+1/2}$     # leading state query
6:     set $X_{t+1} = P_{X_t}(-\gamma_t V_{t+1/2})$       # base state update
7:     set $\beta_t = \dfrac{\|V_{t+1/2} - V_t\|_{X_{t+1/2},*}}{\sqrt{2D(X_{t+1/2}, X_t)}}$       # estimate Bregman constant
8:     set $\gamma_{t+1} = \min\{\gamma_t, \theta\sqrt{K}/\beta_t\}$      # update step-size
9: **end for**

---

successive states in terms of a global norm difference of the form $\|X_{t+1/2} - X_t\|^2$. However, this step also requires $A$ to vary gradually relative to $\|\cdot\|$, which is of course impossible if $A$ exhibits singularities. The key novelty in our setting is the use of the Bregman divergence as a comparator for the algorithm's *successive* states: it is at this point that the triple interplay between the operator, the local norm and the chosen Bregman function is made manifest, and it is what makes Bregman continuity particularly well-suited for tackling singular problems of this kind. This requires a careful treatment of the various Bregman differences involved, so we defer the details to the supplement.

## 5   The adaptive mirror-prox algorithm

A crucial assumption underlying the analysis of the previous section is that the optimizer must know in advance – or be otherwise able to estimate – the Bregman constant $\beta$. In practice, this can be difficult to achieve, so it is important to be able to run (MP) with an *adaptive* step-size policy. Our starting point is the observation that, with perfect oracle feedback, one can estimate $\beta$ by setting

$$\beta_t = \frac{\|A(X_{t+1/2}) - A(X_t)\|_{X_{t+1/2},*}}{\sqrt{2D(X_{t+1/2}, X_t)}} \tag{5.1}$$

whenever $X_{t+1/2} \neq X_t$; obviously, if $A$ is $\beta$-Bregman continuous, we have $\beta_t \leq \beta$.[7] However, the fact that the Bregman constant is being *under*-estimated means that a step-size policy of the form $\gamma_t \propto \sqrt{K}/\beta_t$ would *over*-estimate the inverse Bregman constant $1/\beta$, so the resulting step-size policy would have no reason to satisfy (4.5).

To overcome this obstacle, we introduce the following comparison mechanism: first, at each $t = 1, 2, \ldots$, we use the estimation (5.1) to test the step-size $\bar{\gamma}_t = \sqrt{K}/\beta_t$. Then, to avoid the growth phenomenon outlined above, we shrink $\bar{\gamma}_t$ by a constant factor of $\theta$ and, to avoid running into vanishing step-size issues, we take the previous step-size employed if the shrunk one would be smaller. Formally, we consider the adaptive step-size policy:

$$\gamma_{t+1} = \begin{cases} \min\{\gamma_t, \theta\sqrt{K}/\beta_t\} & \text{if } X_t \neq X_{t+1/2}, \\ \gamma_t & \text{otherwise,} \end{cases} \tag{5.2}$$

with $\beta_t$ defined as in (5.1) and $\theta \in (0, 1)$ chosen arbitrarily.

For concreteness, we call the resulting algorithm *adaptive mirror-prox* (AMP) and we provide a pseudocode implementation in Algorithm 1 above. In terms of performance, we have:

**Theorem 2.** *Assume that $A$ satisfies Assumptions 1 and 2, and (MP) is run with perfect oracle feedback and the adaptive step-size policy (5.2). Then, with notation as in Theorem 1, the algorithm's ergodic average $\bar{X}_T = \sum_{t=1}^{T} \gamma_t X_{t+1/2} / \sum_{t=1}^{T} \gamma_t$ enjoys the gap bound $\mathrm{Gap}_H(\bar{X}_T) = \mathcal{O}(1/T)$.*

We find this result particularly appealing because it yields the optimal $\mathcal{O}(1/T)$ convergence rate of the mirror-prox algorithm, even for possibly singular operators, and even if the operator's Bregman constant is unknown. Its proof relies on using the specific form of the step-size policy (5.2) to control the second term in the energy inequality (4.8); we provide the detailed arguments in the supplement.

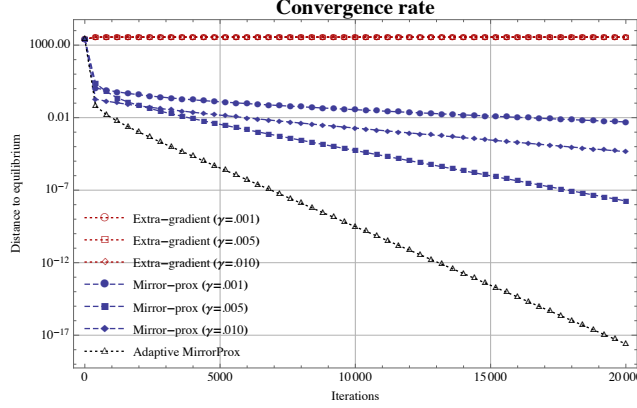

**Figure 1:** Different variants of the mirror-prox algorithm in the resource sharing problem of Example 2.3. The algorithm labeled "extra-gradient" refers to Euclidean regularization and a constant step size as indicated in the legend; "mirror-prox" was run with the Bregman function of Example 3.4 and step-sizes as in the legend; finally, "adaptive mirror-prox" corresponds to Algorithm 1, i.e., mirror-prox with the adaptive step-size policy (5.2).

## 6 Numerical experiments

We performed a series of numerical experiments on the resource sharing problem described in Example 2.3 with a set of $R = 1000$ servers being shared by $N = 100$ commodities, each with a demand drawn uniformly at random from $[0, 1]$; the capacity $c_r$ of each server $r = 1, \ldots, R$ was also drawn randomly from $[0, 100]$. Subsequently, we ran two variants of the mirror-prox method: (MP) with Euclidean regularization, and (MP) with the Bregman function defined in Example 3.4. For all methods, we ran a range of different constant step-sizes (we present the most representative values, namely $\gamma = 0.001$, $\gamma = 0.005$, and $\gamma = 0.010$). Subsequently, we also ran Algorithm 1 and we plotted the distance from the solution to the induced variational inequality problem as a function of the number of iterations. The main conclusions that can be drawn are as follows:

1. The Euclidean version of the mirror-prox algorithm (i.e., the extra-gradient algorithm) is unstable and does not converge; this is due to the fact that the gradients received are very large (recall that the problem is *not* Lipschitz continuous), so the algorithm does not exhibit descent or convergence.

2. The MP variant with the non-Euclidean regularizer of Example 3.4 is convergent (since the VI problem under study is Bregman continuous relative to this Bregman function). However, depending on the method's step-size, the convergence is relatively slow, and there is no easy way to estimate the problem's Bregman constant in order to choose a "good" step-size.

3. By contrast, the AMP algorithm converges significantly faster than variants with a constant step-size. This is due to the fact that, initially, a greedier step-size is able to take larger steps towards the problem's solution, so initializing Algorithm 1 with a large step-size helps significantly.

## 7 Concluding remarks

In this work, we introduced a novel regularity condition to account for variational inequalities (both deterministic and stochastic) with possible singularities. This condition, which we call *Bregman continuity*, is tailored to the operator's singularity landscape and, as such, provides the necessary bedrock to achieve optimal convergence rates via a properly chosen version of the mirror-prox algorithm (with or without knowledge of the problem's Bregman constant). This opens up several interesting research directions: First, an appealing extension would be to develop a "model-agnostic" version of the method (which would concurrently provide optimal rates in stochastic and deterministic settings) or to combine it with backtracking / linesearch to accelerate convergence. Finally, it would also be interesting to examine the method's local convergence properties in non-monotone problems (deterministic or stochastic). We relegate these questions to future work.

## Acknowledgments

The authors gratefully acknowledge financial support from the French National Research Agency (ANR) under grants ORACLESS (ANR–16–CE33–0004–01) and ELIOT (ANR-18-CE40-0030).

## Footnotes

[1]When used to find a zero of a composite operator, the FB algorithm is known as a "splitting" method; see e.g., Bruck Jr. [12], Passty [44], [14], and references therein.

[2]An operator $A(x)$ is cocoercive if $\langle A(x') - A(x), x' - x \rangle \geq (1/\beta)\|A(x') - A(x)\|^2$ for some $\beta > 0$ and all $x, x'$. Note that Lipschitz continuity is strictly weaker than cocoercivity: the operator $A(x_1, x_2) = (-x_2, x_1)$ is Lipschitz continuous over $\mathbb{R}^2$, but it is not cocoercive; see Section 2 for a detailed discussion.

[3]We note here that the method is sometimes referred to as "optimistic mirror descent". This terminology is due to Rakhlin and Sridharan [45, 46] and may refer either to the mirror-prox method itself, or to a variant with "gradient extrapolation from the past", as in [17].

[4]For posterity, note here that $\mathcal{X}$ is convex but it is not necessarily closed.

[5]Recall here that the dual norm of $v \in \mathcal{V}^*$ is defined as $\|v\|_* = \max_{z \in \mathcal{V}}\{|\langle v, z \rangle| : \|z\| \le 1\}$.

[6]By that, we have in mind the definition of an absolutely homogeneous Finsler metric [5]. Specifically, a local norm is viewed here as continuous nonnegative function $F\colon \mathcal{X} \times \mathcal{V} \to \mathbb{R}_+$ with the following propoerties: for all $x \in \mathcal{X}$ and all $z_1, z_2 \in \mathcal{V}$, we have (i) $F(x, z_1 + z_2) \leq F(x, z_1) + F(x, z_2)$; (ii) $F(x, \lambda z) = |\lambda|z$; and (iii) $F(x, z) > 0$ for all $z \in \mathcal{V} \setminus \{0\}$. The local norm of $z$ at $x$ is then defined as $\|z\|_x = F(x, z)$.

[7]In a Euclidean setting, similar ideas can be found in, e.g., [8, 29]. We ignore the origins of this technique.

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
