[Supplementary Material]

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

# A    Properties of the restricted gap function

In this appendix, we discuss the basic properites of the restricted merit function $\mathrm{Gap}_{\mathcal{C}}$ introduced in the main text. For completeness, we begin with the proof of Lemma 1, itself an extension of a similar result by Nesterov (2007):

*Proof of Lemma 1.* Let $x^* \in \mathcal{X}$ be a solution of (VI) so $\langle A(x^*), x - x^* \rangle \geq 0$ for all $x \in \mathcal{X}$. Then, by monotonicity, we get:

$$\langle A(x), x^* - x \rangle \leq \langle A(x) - A(x^*), x^* - x \rangle + \langle A(x^*), x^* - x \rangle$$
$$= -\langle A(x^*) - A(x), x^* - x \rangle - \langle A(x^*), x - x^* \rangle \leq 0, \tag{A.1}$$

so $\mathrm{Gap}_{\mathcal{C}}(x^*) \leq 0$. On the other hand, if $x^* \in \mathcal{C}$, we also get $\mathrm{Gap}(x^*) \geq \langle A(x^*), x^* - x^* \rangle = 0$, so we conclude that $\mathrm{Gap}_{\mathcal{C}}(x^*) = 0$.

For the converse statement, assume that $\mathrm{Gap}_{\mathcal{C}}(\hat{x}) = 0$ for some $\hat{x} \in \mathcal{C}$ and suppose that $\mathcal{C}$ contains a neighborhood of $\hat{x}$ in $\mathcal{X}$. First, we claim that the following inequality holds:

$$\langle A(x), x - \hat{x} \rangle \geq 0 \quad \text{for all } x \in \mathcal{C}. \tag{A.2}$$

Indeed, assume to the contrary that there exists some $x_1 \in \mathcal{C}$ such that

$$\langle A(x_1), x_1 - \hat{x} \rangle < 0. \tag{A.3}$$

This would then give

$$0 = \mathrm{Gap}_{\mathcal{C}}(\hat{x}) \geq \langle A(x_1), \hat{x} - x_1 \rangle > 0, \tag{A.4}$$

which is a contradiction. Now, we further claim that $\hat{x}$ is a solution of (VI),i.e.,:

$$\langle A(\hat{x}), x - \hat{x} \rangle \geq 0 \text{ for all } x \in \mathcal{X}. \tag{A.5}$$

If we suppose that there exists some $z_1 \in \mathcal{X}$ such that $\langle A(\hat{x}), z_1 - \hat{x} \rangle < 0$, then, by the continuity of $A$, there exists a neighborhood $U'$ of $\hat{x}$ in $\mathcal{X}$ such that

$$\langle A(x), z_1 - x \rangle < 0 \quad \text{for all } x \in U'. \tag{A.6}$$

Hence, assuming without loss of generality that $U' \subset U \subset \mathcal{C}$ (the latter assumption due to the assumption that $\mathcal{C}$ contains a neighborhood of $\hat{x}$), and taking $\lambda > 0$ sufficiently small so that $x = \hat{x} + \lambda(z_1 - \hat{x}) \in U'$, we get that $\langle A(x), x - \hat{x} \rangle = \lambda \langle A(x), z_1 - \hat{x} \rangle < 0$, in contradiction to (A.2). We conclude that $\hat{x}$ is a solution of (VI), as claimed. $\qquad\square$

For intuition, we discuss below the relation of the error function $\mathrm{Gap} \equiv \mathrm{Gap}_{\mathcal{X}}$ to other performance measures that arise in practice (similar considerations also apply to $\mathrm{Gap}_{\mathcal{C}}$ as well):

(1) If $A = \nabla f$ for a convex differentiable function $f$ with $\inf_{x \in \mathcal{X}} f(x) > -\infty$, we have, for all $x \in \mathcal{X}$, $\langle \nabla f(x), \hat{x} - x \rangle \leq f(\hat{x}) - f(x)$, so $\mathrm{Gap}(\hat{x}) \leq \sup_x [f(\hat{x}) - f(x)] \leq f(\hat{x}) - \inf f$.

(2) In saddle-point problems, the quality of a candidate solution $\hat{x} = (\hat{x}_1, \hat{x}_2)$ is often assessed via the *Nikaido–Isoda* (NI) function

$$\mathrm{NI}(\hat{x}) = \sup_{x_2 \in \mathcal{X}_2} f(\hat{x}_1, x_2) - \inf_{x_1 \in \mathcal{X}_1} f(x_1, \hat{x}_2) \tag{NI}$$

provided of course that the right-hand side is well-posed. Similarly to the minimization framework above, if $f$ is convex-concave, we have $\mathrm{Gap}(\hat{x}) \leq \mathrm{NI}(\hat{x})$.

In view of all this, a bound on $\mathrm{Gap}(\hat{x})$ does not immediately translate to a bound on the value of the loss or the Nikaido–Isoda function (for minimization or min-max problems respectively). However, the same arguments used to obtain the convergence rate of an algorithm relative to $\mathrm{Gap}$ can be usually adapted to these measures with minimal effort.

# B Bregman functions and Bregman continuity

## B.1 Properties of Bregman functions

In this appendix, we present some basic facts about Bregman functions and proximal mappings. Similar results exist in the literature in different contexts (see e.g., [22, 41, 42] and references therein), but given that many of our results rely on the use of *local* – as opposed to *global* – norms, we provide here complete statements and proofs.

To begin, we introduce two notions that will be particularly useful in the sequel. The first is the convex conjugate of a Bregman function $h$, i.e.,

$$h^*(y) = \max_{x \in \mathcal{V}}\{\langle y, x \rangle - h(x)\} \tag{B.1}$$

and the associated primal-dual "mirror map" $Q: \mathcal{V}^* \to \mathcal{X}$:

$$Q(y) = \arg\max_{x \in \mathcal{V}}\{\langle y, x \rangle - h(x)\} \tag{B.2}$$

We then have the following basic lemma connecting the above notions:

**Lemma B.1.** *Let $h$ be a $K$-strongly convex Bregman function as above. Then, for all $x \in \mathcal{X}^\circ \equiv \mathrm{dom}\,\partial h$ and all $v, y \in \mathcal{V}^*$ we have:*

1. $x = Q(y) \iff y \in \partial h(x)$

2. $x^+ = P_x(v) \iff \nabla h(x) + v \in \partial h(x^+) \iff x^+ = Q(\nabla(x) + v)$

3. *Finally, if $x = Q(y)$ and $p \in \mathcal{X}$, we get:*

$$\langle \nabla h(x), x - p \rangle \le \langle y, x - p \rangle \tag{B.3}$$

*Proof.* For the first equivalence, note that $x$ solves (B.1) if and only if $0 \in y - \partial h(x)$ and hence if and only if $y \in \partial h(x)$. Working in the same spirit for the second equivalence, we have that $x^+$ solves (4.3) if and only if $\nabla h(x) + v \in \partial h(x^+)$ and therefore if and only if $x^+ = Q(\nabla h(x) + v)$.

For our last claim, by a simple continuity argument, it is sufficient to show that the inequality holds for the relative interior $\mathrm{ri}\,\mathcal{X}$ of $\mathcal{X}$. In order to show this, pick a base point $p \in \mathrm{ri}\,\mathcal{X}$, and let

$$\phi(t) = h(x + t(p - x)) - [h(x) + \langle y, x + t(p - x) \rangle] \quad \text{for all } t \in [0, 1]. \tag{B.4}$$

Since, $h$ is strongly convex and $y \in \partial h(x)$ due to the first equivalence, it follows that $\phi(t) \ge 0$ with equality if and only if $t = 0$. Since, $\psi(t) = \langle \nabla h(x + t(p - x)) - y, p - x \rangle$ is a continuous selection of subgradients of $\phi$ and both $\phi$ and $\psi$ are continuous over $[0, 1]$, it follows that $\phi$ is continuously differentiable with $\phi' = \psi$ on $[0, 1]$. Hence, with $\phi$ convex and $\phi(t) \ge 0 = \phi(0)$ for all $t \in [0, 1]$, we conclude that $\phi'(0) = \langle \nabla h(x) - y, p - x \rangle \ge 0$ and thus we obtain the result. $\square$

The basic ingredient for establishing connections in the Bregman framework is a generalization of the rule of cosines which is known in the literature as the "three-point identity" [14] and will be the main tool for deriving the main estimations for our analysis. Being more precise, we have the following lemma:

**Lemma B.2.** *Let $h$ be a Bregman function on $\mathcal{X}$. Then, for all $p \in \mathcal{X}$ and all $x, x' \in \mathcal{X}^\circ$, we have:*

$$D(p, x') = D(p, x) + D(x, x') + \langle \nabla h(x') - \nabla h(x), x - p \rangle \tag{B.5}$$

The proof of this lemma follows as in the classic Bregman case [14] so we omit it and proceed to derive some key bounds for the Bregman divergence before and after a mirror step:

**Proposition B.1.** *Let $h$ be a local Bregman function with strong convexity modulus $K > 0$. Fix some $p \in \mathcal{X}$ and let $x^+ = P_x(v)$ for some $x \in \mathcal{X}^\circ$ and $v \in \mathcal{V}^*$. We then have:*

$$D(p, x^+) \le D(p, x) - D(x^+, x) + \langle v, x^+ - p \rangle \tag{B.6}$$

*Proof.* By the three-point identity established in Lemma B.2, we get:

$$D(p,x) = D(p,x^+) + D(x^+,x) + \langle \nabla h(x) - \nabla h(x^+), x^+ - p \rangle \tag{B.7}$$

By rearranging the terms we get:

$$D(p,x^+) = D(p,x) - D(x^+,x) + \langle \nabla h(x^+) - \nabla h(x), x^+ - p \rangle \tag{B.8}$$

Due to (B.3) and the fact that $x^+ = P_x(v)$ so $\nabla h(x) + v \in \partial h(x^+)$, we get the result. $\square$

Thanks to the above estimations, we obtain the following inequalities relating the Bregman divergence between *two* prox-steps:

**Proposition B.2.** *Let $h$ be a Bregman function on $\mathcal{X}$ and fix some $p \in \mathcal{X}$, $x \in \mathcal{X}^\circ$. Letting $x_1^+ = P_x(v_1)$ and $x_2^+ = P_x(v_2)$, we have:*

$$D(p,x_2^+) \le D(p,x) + \langle v_2, x_1^+ - p \rangle + [\langle v_2, x_2^+ - x_1^+ \rangle - D(x_2^+,x)] \tag{B.9a}$$

$$\le D(p,x) + \langle v_2, x_1^+ - p \rangle + \langle v_2 - v_1, x_2^+ - x_1^+ \rangle - D(x_2^+,x_1^+) - D(x_1^+,x). \tag{B.9b}$$

*Proof.* For the first inequality, by applying Proposition B.1 for $x_2^+ = P_x(v_2)$, we get:

$$\begin{aligned} D(p,x_2^+) &\le D(p,x) - D(x_2^+,x) + \langle v_2, x_2^+ - p \rangle \\ &= D(p,x) + \langle v_2, x_1^+ - p \rangle + [\langle v_2, x_2^+ - x_1^+ \rangle - D(x_2^+,x)] \end{aligned} \tag{B.10}$$

For the second inequality, we need to bound $\langle v_2, x_2^+ - x_1^+ \rangle - D_h(x_2^+,x)$. In particular, applying again Proposition B.1 for $p = x_2^+$, we get:

$$D(x_2^+,x_1^+) \le D(x_2^+,x) + \langle v_1, x_1^+ - x_2^+ \rangle - D(x_1^+,x) \tag{B.11}$$

and hence:

$$D(x_2^+,x) \ge D(x_2^+,x_1^+) + D(x_1^+,x) - \langle v_1, x_1^+ - x_2^+ \rangle. \tag{B.12}$$

So, combining the above inequalities we get:

$$\langle v_2, x_2^+ - x_1^+ \rangle - D(x_2^+,x) \le \langle v_2, x_2^+ - x_1^+ \rangle - D(x_2^+,x_1^+) - D(x_1^+,x) - \langle v_1, x_2^+ - x_1^+ \rangle \tag{B.13}$$

and thus we get the second inequality as well. $\square$

## B.2 Bregman cocoercivity

For comparison to the operator theory literature, we end this section by introducing a notion close to Bregman continuity, that of *Bregman cocoercivity*. In particular, we have the following definition:

**Definition 4.** *Let $A\colon \mathcal{X} \to \mathcal{V}^*$ be an operator, let $\|\cdot\|_x$ be a local norm on $\mathcal{X}$ and let $h$ be a local Bregman function on $\mathcal{X}$ with strong convexity modulus $K$. We say that $A$ is $\delta$-Bregman cocoercive if*

$$\langle A(x) - A(x'), x - x' \rangle \ge \delta \|A(x) - A(x')\|_{*,x} \|A(x) - A(x')\|_{*,x'} \quad \text{for all } x,x' \in \mathcal{X}. \tag{B.14}$$

This notion is a straightforward extension of ordinary operator cocoercivity, e.g., as defined in [6]. As in the base case, we have:

**Lemma B.3.** *Let $h$ be a Bregman function with local strong convexity modulus $K > 0$ and let $A\colon \mathcal{X} \to \mathcal{V}^*$ be a $\delta$-Bregman cocoercive operator. Then, $A$ is $1/(\delta\sqrt{K})$-Bregman continuous.*

*Proof.* By applying the Cauchy-Shwartz inequality to the definition (B.14) of Bregman cocoercivity, we get:

$$\begin{aligned} \delta \|A(x) - A(x')\|_{*,x} \|A(x) - A(x')\|_{*,x'} &\le \langle A(x') - A(x), x' - x \rangle \\ &\le \|A(x') - A(x)\|_{*,x'} \|x' - x\|_{x'} \end{aligned} \tag{B.15}$$

Thus, by simplifying and recalling Lemma 2 in the main paper, we get:

$$\|A(x) - A(x')\|_{*,x} \le \frac{1}{\delta}\|x - x'\|_{x'} \le \frac{1}{\delta\sqrt{K}}\sqrt{2D(x,x')} \tag{B.16}$$

i.e., $A$ is Bregman continuous with modulus of continuity $1/(\delta\sqrt{K})$, as claimed. $\square$

## C  Convergence analysis of the mirror-prox algorithm

In this section, we provide the proofs of our main convergence results for the mirror-prox algorithm as presented in Sections 4 and 5 of the main part of the paper.

### C.1  Deterministic analysis

The main ingredient of our proof for the deterministic case is the following energy inequality:

**Proposition C.1.** *Assume that A satisfies Assumption 2 and* (MP) *is run with perfect oracle feedback. Then, for all $p \in \mathcal{X}$, we have:*

$$D(p, X_{t+1}) \le D(p, X_t) - \gamma_t \langle A(X_{t+\frac{1}{2}}), X_{t+\frac{1}{2}} - p \rangle - \left(1 - \frac{\beta^2 \gamma_t^2}{K}\right) D(X_{t+\frac{1}{2}}, X_t).$$

*Proof.* By setting $x = X_t$, $y_1 = -\gamma_t A(X_t)$, $x_1^+ = X_{t+\frac{1}{2}}$, $y_2 = -\gamma_t A(X_{t+\frac{1}{2}})$ and $x_2^+ = X_{t+1}$ in Proposition B.2, we readily obtain:

$$\begin{aligned}
D(p, X_{t+1}) \le{}& D(p, X_t) - \gamma_t \langle A(X_{t+\frac{1}{2}}), X_{t+\frac{1}{2}} - p \rangle \\
& - \gamma_t \langle A(X_{t+\frac{1}{2}}) - A(X_t), X_{t+1} - X_{t+\frac{1}{2}} \rangle \\
& - D(X_{t+1}, X_{t+\frac{1}{2}}) - D(X_{t+\frac{1}{2}}, X_t).
\end{aligned} \tag{C.1}$$

Proceeding line-by-line, the Fenchel-Young inequality applied to the function $\phi(x) = \|x\|_{X_{t+\frac{1}{2}}}^2$ further gives

$$\begin{aligned}
\langle A(X_{t+\frac{1}{2}}) - A(X_t), X_{t+1} - X_{t+\frac{1}{2}} \rangle \le{}& \frac{K}{2\gamma_t} \|X_{t+1} - X_{t+\frac{1}{2}}\|_{X_{t+\frac{1}{2}}}^2 \\
& + \frac{\gamma_t}{2K} \|A(X_{t+\frac{1}{2}}) - A(X_t)\|_{X_{t+\frac{1}{2}}, *}.
\end{aligned} \tag{C.2}$$

Thus, by substituting in (C.1), we get

$$\begin{aligned}
D(p, X_{t+1}) \le{}& D(p, X_t) - \gamma_t \langle A(X_{t+\frac{1}{2}}), X_{t+\frac{1}{2}} - p \rangle \\
& + \frac{K}{2} \|X_{t+1} - X_{t+\frac{1}{2}}\|_{X_{t+\frac{1}{2}}}^2 + \frac{\gamma_t^2}{2K} \|A(X_{t+\frac{1}{2}}) - A(X_t)\|_{X_{t+\frac{1}{2}}, *}^2 \\
& - D(X_{t+1}, X_{t+\frac{1}{2}}) - D(X_{t+\frac{1}{2}}, X_t).
\end{aligned} \tag{C.3}$$

and hence, by Lemma 2, we obtain:

$$\begin{aligned}
D(p, X_{t+1}) \le{}& D(p, X_t) - \gamma_t \langle A(X_{t+\frac{1}{2}}), X_{t+\frac{1}{2}} - p \rangle \\
& + \frac{\gamma_t^2}{2K} \|A(X_{t+\frac{1}{2}}) - A(X_t)\|_{X_{t+\frac{1}{2}}, *}^2 - D(X_{t+\frac{1}{2}}, X_t).
\end{aligned} \tag{C.4}$$

However, the Bregman continuity of $A$ also yields

$$\|A(X_{t+\frac{1}{2}}) - A(X_t)\|_{X_{t+\frac{1}{2}}, *}^2 \le 2\beta^2 D(X_{t+\frac{1}{2}}, X_t) \tag{C.5}$$

so our claim follows by combining Eqs. (C.4) and (C.5).  $\square$

We are now in a position to establish the $\mathcal{O}(1/T)$ convergence rate of (MP) for deterministic problems:

*Proof of Theorem 1 - deterministic case.* Fix some $p \in \mathcal{C}_H$. Since $\gamma_t \le 1/\beta$ by assumption, a slight rearrangement of Proposition C.1 readily yields:

$$\gamma_t \langle A(X_{t+\frac{1}{2}}), X_{t+\frac{1}{2}} - p \rangle \le D(p, X_t) - D(p, X_{t+1}) \tag{C.6}$$

Moreover, by the monotonicity of $A$, we also have:

$$\langle A(p), X_{t+\frac{1}{2}} - p \rangle \le \langle A(X_{t+\frac{1}{2}}), X_{t+\frac{1}{2}} - p \rangle. \tag{C.7}$$

Thus, combining the two inequalities above, we get

$$\gamma_t \langle A(p), X_{t+\frac{1}{2}} - p \rangle \leq D(p, X_t) - D(p, X_{t+1}) \tag{C.8}$$

and, proceeding to telescope from $t = 1$ to $T$, we obtain:

$$\sum_{t=1}^{T} \gamma_t \langle A(p), X_{t+\frac{1}{2}} - p \rangle \leq D(p, X_1) - D(p, X_{t+1}) \leq D(p, x_c) \tag{C.9}$$

Then, dividing by $\sum_{t=1}^{T} \gamma_t$ finally yields

$$\langle A(p), \bar{X}_T - p \rangle \leq \frac{D(p, x_c)}{\sum_{t=1}^{T} \gamma_t} \leq \frac{D(p, x_c)}{\gamma_{\min} T}, \tag{C.10}$$

so our result follows by taking the supremum over all $p \in \mathcal{X}$ such that $D(p, x_c) \leq H$ (i.e., over all $p \in \mathcal{C}_H$). □

## C.2 Stochastic analysis

We now turn to our results for the convergence of the mirror-prox algorithm in the stochastic case:

*Proof of Theorem 1 - stochastic case.* Working in the same spirit as for the deterministic case, let $x = X_t$, $y_1 = -\gamma_t V_t$, $x_1^+ = X_{t+\frac{1}{2}}$, $y_2 = -\gamma_t V_{t+\frac{1}{2}}$ and $x_2^+ = X_{t+1}$ in the first part of Proposition B.2. We then get:

$$\begin{aligned} D(p, X_{t+1}) &\leq D(p, X_t) - \gamma_t \langle V_{t+\frac{1}{2}}, X_{t+\frac{1}{2}} - p \rangle \\ &\quad + \left[ \gamma_t \langle V_{t+\frac{1}{2}}, X_{t+1} - X_{t+\frac{1}{2}} \rangle - D(X_{t+1}, X_t) \right] \\ &\leq D(p, X_t) - \gamma_t \langle A(X_{t+\frac{1}{2}}), X_{t+\frac{1}{2}} - p \rangle \\ &\quad - \gamma_t \xi_{t+\frac{1}{2}} + \left[ \gamma_t \langle V_{t+\frac{1}{2}}, X_{t+1} - X_{t+\frac{1}{2}} \rangle - D(X_{t+1}, X_t) \right] \end{aligned} \tag{C.11}$$

where we used the feedback decomposition $V_{t+\frac{1}{2}} = A(X_{t+\frac{1}{2}}) + U_{t+\frac{1}{2}}$ for $V_{t+\frac{1}{2}}$ and we set $\xi_{t+\frac{1}{2}} = \langle U_{t+\frac{1}{2}}, X_{t+\frac{1}{2}} - p \rangle$ in the last line. By the second part of Proposition B.2, we also have

$$\begin{aligned} \gamma_t \langle V_{t+\frac{1}{2}}, X_{t+1} - X_{t+\frac{1}{2}} \rangle - D(X_{t+1}, X_t) &\leq \gamma_t \langle V_t - V_{t+\frac{1}{2}}, X_{t+1} - X_{t+\frac{1}{2}} \rangle \\ &\quad - D(X_{t+1}, X_{t+\frac{1}{2}}) - D(X_{t+\frac{1}{2}}, X_t) \end{aligned} \tag{C.12}$$

Now, by applying the Fenchel-Young inequality to the duality pairing in the above inequality, we get

$$\gamma_t \langle V_t - V_{t+\frac{1}{2}}, X_{t+1} - X_{t+\frac{1}{2}} \rangle \leq \frac{\gamma_t^2}{2K} \|V_t - V_{t+\frac{1}{2}}\|_{X_{t+\frac{1}{2}},*}^2 + \frac{K}{2} \|X_{t+1} - X_{t+\frac{1}{2}}\|_{X_{t+\frac{1}{2}}}^2. \tag{C.13}$$

On the other hand, by the stochastic oracle assumption (4.1), we have:

$$\begin{aligned} \frac{\gamma_t^2}{2K} \|V_t - V_{t+\frac{1}{2}}\|_{X_{t+\frac{1}{2}},*}^2 &\leq \frac{\gamma_t^2}{K} \|A(X_t) - A(X_{t+\frac{1}{2}})\|_{X_{t+\frac{1}{2}},*}^2 + \frac{\gamma_t^2}{K} \|U_t - U_{t+\frac{1}{2}}\|_{X_{t+\frac{1}{2}},*}^2 \\ &\leq \frac{2\beta^2 \gamma_t^2}{K} D(X_{t+\frac{1}{2}}, X_t) + \frac{\gamma_t^2}{\mu K} \|U_t - U_{t+\frac{1}{2}}\|_*^2. \end{aligned} \tag{C.14}$$

where the last line follows from the Bregman continuity of $A$ (Assumption 2) and the fact that $\|\cdot\|_x \geq \mu \|\cdot\|$ for some $\mu > 0$ and all $x \in \mathcal{X}$ (implying in turn that $\|\cdot\|_{x,*} \leq \mu^{-1} \|\cdot\|_*$ for all $x \in \mathcal{X}$). We thus get

$$\gamma_t \langle V_{t+\frac{1}{2}}, X_{t+1} - X_{t+\frac{1}{2}} \rangle - D(X_{t+1}, X_t) \leq \left( \frac{2\beta^2 \gamma_t^2}{K} - 1 \right) D(X_{t+\frac{1}{2}}, X_t) + \frac{\gamma_t^2}{\mu K} \|U_t - U_{t+\frac{1}{2}}\|_*^2 \tag{C.15}$$

Since $\gamma_t^2 \leq K/(2\beta^2)$ by assumption, substituting (C.15) in (C.11) and rearranging yields

$$\gamma_t \langle A(X_{t+\frac{1}{2}}), X_{t+\frac{1}{2}} - p \rangle \leq D(p, X_t) - D(p, X_{t+1}) - \gamma_t \xi_{t+\frac{1}{2}} + \frac{\gamma_t^2}{\mu K} \|U_t - U_{t+\frac{1}{2}}\|_*^2$$

$$\leq D(p, X_t) - D(p, X_{t+1})) - \gamma_t \xi_{t+\frac{1}{2}} + \frac{2\gamma_t^2}{\mu K} \Big[ \|U_t\|_*^2 + \|U_{t+\frac{1}{2}}\|_*^2 \Big].$$

(C.16)

In order to bound $\xi_{t+\frac{1}{2}}$, we will need to introduce the auxilliary process

$$Z_{t+1} = \underset{x \in \mathcal{X}}{\arg\min} \{ \langle U_{t+\frac{1}{2}}, Z_t - x \rangle + \frac{\mu}{\gamma_t} D(x, Z_t) \}$$

(C.17)

with $Z_1 = x_c$. We then have

$$-\gamma_t \xi_{t+1} = \gamma_t \langle U_{t+\frac{1}{2}}, p - X_{t+\frac{1}{2}} \rangle = \gamma_t \langle U_{t+\frac{1}{2}}, Z_t - X_{t+\frac{1}{2}} \rangle + \gamma_t \langle U_{t+\frac{1}{2}}, p - Z_t \rangle$$

(C.18)

In order to bound the term which depends on $p$, we have the following:

$$\gamma_t \langle U_{t+\frac{1}{2}}, p - Z_t \rangle = \gamma_t \langle U_{t+\frac{1}{2}}, p - Z_{t+1} \rangle + \gamma_t \langle U_{t+\frac{1}{2}}, Z_{t+1} - Z_t \rangle$$

$$\leq \mu \langle \nabla h(Z_{t+1}) - \nabla h(Z_t), p - Z_{t+1} \rangle + \frac{\gamma_t^2}{2K} \|U_{t+\frac{1}{2}}\|_{*, Z_t}^2 + \frac{K}{2} \|Z_{t+1} - Z_t\|_{Z_t}^2$$

$$\leq \mu \langle \nabla h(Z_{t+1}) - \nabla h(Z_t), p - Z_{t+1} \rangle + \frac{\gamma_t^2}{2\mu K} \|U_{t+\frac{1}{2}}\|_*^2 + \frac{K\mu}{2} \|Z_{t+1} - Z_t\|^2.$$

(C.19)

Hence, by the three-point identity, we obtain:

$$\gamma_t \langle U_{t+\frac{1}{2}}, p - Z_t \rangle \leq \mu [D(p, Z_t) - D(p, Z_{t+1})] - \mu D(Z_{t+1}, Z_t)$$

$$+ \frac{\gamma_t^2}{2\mu K} \|U_{t+\frac{1}{2}}\|_*^2 + \frac{K\mu}{2} \|Z_{t+1} - Z_t\|^2$$

$$\leq \mu [D(p, Z_t) - D(p, Z_{t+1})] + \frac{\gamma_t^2}{2\mu K} \|U_{t+\frac{1}{2}}\|_*^2$$

(C.20)

where the last inequality is a consequence of the strong convexity of $h$. Thus, combining all these with the fact that $A$ is monotone, we can telescope and obtain

$$\sum_{t=1}^{T} \gamma_t \langle A(p), X_{t+\frac{1}{2}} - p \rangle \leq (1 + \mu) D(p, x_c)$$

$$+ \sum_{t=1}^{T} \gamma_t \langle U_{t+\frac{1}{2}}, Z_t - X_{t+\frac{1}{2}} \rangle$$

$$+ \frac{1}{\mu K} \sum_{t=1}^{T} \gamma_t^2 \Big[ 2\|U_t\|_*^2 + \frac{5}{2} \|U_{t+\frac{1}{2}}\|_*^2 \Big].$$

Hence, after dividing by $\sum_{t=1}^{T} \gamma_t$ and taking the supremum over $p \in \mathcal{C}_H$, we get:

$$\text{Gap}_H(\bar{X}_T) \leq \frac{(1 + \mu) H + \sum_{t=1}^{T} \gamma_t \langle U_{t+\frac{1}{2}}, Z_t - X_{t+\frac{1}{2}} \rangle + \frac{1}{\mu K} \sum_{t=1}^{T} \gamma_t^2 \Big[ 2\|U_t\|_*^2 + \frac{5}{2} \|U_{t+\frac{1}{2}}\|_*^2 \Big]}{\sum_{t=1}^{T} \gamma_t}.$$

(C.21)

Since $\mathbb{E}[\langle U_{t+\frac{1}{2}}, Z_t - X_{t+\frac{1}{2}} \rangle] = \mathbb{E}[\mathbb{E}[\langle U_{t+\frac{1}{2}}, Z_t - X_{t+\frac{1}{2}} \rangle] \mid \mathcal{F}_{t+\frac{1}{2}}] = 0$, taking expectations yields

$$\mathbb{E}[\text{Gap}_H(\bar{X}_t)] \leq \frac{(1 + \mu) D + \frac{9\sigma^2}{2\mu K} \sum_{t=1}^{T} \gamma_t^2}{\sum_{t=1}^{T} \gamma_t},$$

(C.22)

which proves our claim. Finally, the RHS of this last inequality is $\tilde{\mathcal{O}}(1/T^{1/2})$ if $\gamma_t \propto 1/\sqrt{t}$, so the $\tilde{\mathcal{O}}(1/\sqrt{T})$ result follows. □

## C.3 Adaptive analysis

Our aim here is to prove the $\mathcal{O}(1/T)$ convergence rate of the adaptive mirror-prox algorithm:

*Proof of Theorem 2.* For simplicity, we will assume in what follows that $X_{t+\frac{1}{2}} \neq X_t$ for all $t \geq 1$. Otherwise, if $X_{t+\frac{1}{2}} = X_t$ for some $t$, we would also have $\gamma_{t+1} = \gamma_t$ for said value of $t$ by convention; this would not change our arguments below, but it would make them much more cumbersome to write down.

With this caveat in mind, we begin by showing that the adaptive step-size policy $\gamma_{t+1} = \min\{\gamma_t, \theta\sqrt{K}/\beta_t\}$ is bounded from below as

$$\gamma_t \geq \min\{\gamma_1, \theta\sqrt{K}/\beta\}. \tag{C.23}$$

We consider two cases below: First, if $\gamma_1 \leq \theta\sqrt{K}/\beta$, we will also have

$$\frac{\theta\sqrt{K}}{\beta_t} \geq \frac{\theta\sqrt{K}}{\beta} \geq \gamma_1 \tag{C.24}$$

for all $t \geq 1$. We then claim that $\gamma_t = \gamma_1$ for all $t \geq 1$: indeed, assuming inductively that this is the case for some $\geq\geq 1$ (the claim is trivially true for $t = 1$), we readily get

$$\gamma_{t+1} = \min\{\gamma_t, \theta\sqrt{K}/\beta_t\} = \min\{\gamma_1, \theta\sqrt{K}/\beta_t\} = \gamma_1, \tag{C.25}$$

which proves our claim in this case.

Assume now that $\gamma_1 > \theta\sqrt{K}/\beta$, in which case we are left to show that $\gamma_t \geq \theta\sqrt{K}/\beta$ for all $t \geq 1$. Again, the base case $t = 1$ being true trivially, assume inductively that this holds for some $t \geq 1$. Then, one of the following will hold:

1. $\gamma_{t+1} = \gamma_t$ so $\gamma_{t+1} \geq \theta\sqrt{K}/\beta$ by the inductive assumption.

2. $\gamma_{t+1} = \theta\sqrt{K}/\beta_t \geq \theta\sqrt{K}/\beta$ by the fact that $\beta_t$ is an under-estimate of $\beta$.

In both cases we obtain $\gamma_{t+1} \geq \theta\sqrt{K}/\beta$, so the induction is complete.

To proceed, going back to the proof of the basic energy inequality (C.1), the intermediate step (C.4) can be rewritten as

$$D(p, X_{t+1}) \leq D(p, X_t) - \gamma_t \langle A(X_{t+\frac{1}{2}}), X_{t+\frac{1}{2}} - p \rangle - \left(1 - \frac{\beta_t^2 \gamma_t^2}{K}\right) D(X_{t+\frac{1}{2}}, X_t)$$

$$\leq D(p, X_t) - \gamma_t \langle A(X_{t+\frac{1}{2}}), X_{t+\frac{1}{2}} - p \rangle - \left(1 - \theta^2 \frac{\gamma_t^2}{\gamma_{t+1}^2}\right) D(X_{t+\frac{1}{2}}, X_t), \tag{C.26}$$

where we used the fact that $\gamma_{t+1} \leq \theta\sqrt{K}/\beta_t$ by construction. However, since $\gamma_{t+1}$ is weakly decreasing and bounded from below, it follows that $\gamma_t$ converges to some limit value $\gamma_\infty$ as $t \to \infty$. In turn, this implies that

$$\lim_{t \to \infty} \left(1 - \theta^2 \frac{\gamma_t^2}{\gamma_{t+1}^2}\right) = 1 - \theta^2 > 0, \tag{C.27}$$

and, hence

$$\left(1 - \theta^2 \frac{\gamma_t^2}{\gamma_{t+1}^2}\right) D(X_{t+\frac{1}{2}}, X_t) > 0 \tag{C.28}$$

for all $t$ greater than some (finite) $t_0$. Accordingly, rearranging (C.26) and subsequently telescoping as in (C.9), we finally obtain

$$\sum_{t=1}^{T} \gamma_t \langle A(p), X_{t+\frac{1}{2}} - p \rangle \leq D(p, X_1) - D(p, X_{t+1}) - \sum_{t=1}^{T} \left(1 - \theta^2 \frac{\gamma_t^2}{\gamma_{t+1}^2}\right) D(X_{t+\frac{1}{2}}, X_t)$$

$$\leq D(p, x_c) - \sum_{t=1}^{t_0} \left(1 - \theta^2 \frac{\gamma_t^2}{\gamma_{t+1}^2}\right) D(X_{t+\frac{1}{2}}, X_t) < \infty \tag{C.29}$$

whenever $T > t_0$. Our result then follows by dividing both sides of this last inequality by $\sum_{t=1}^{T} \gamma_t$ and recalling the fact that $\gamma_t \geq \min\{\gamma_1, \theta\sqrt{K}/\beta\} > 0$ for all $t$. $\qquad\square$