[Reviews · NeurIPS 2019]

Reviewer 1



The paper is well-written and is easy to follow, although prior knowledge of the notations and concepts are occasionally required. I did not carefully check the results, but they generally follow a reasonable line of argument.

Reviewer 2



Among the positive aspects of this paper is that it' well written, the theoretical results are clearly explained and the related work review is fair and engaging. Among the negative aspects: 1. It's unclear what is the dual norm in (4.2b), since all previous definitions refer to the local norm || ||_x . If this is not a local norm, then the claims that this analysis extends to an arbitrary domain are void since this variance is typically unbounded for simple functions like quadratics. 2. Part of the contributions of this paper are shared with paper ID #4247, who uses the same framework based on local norms applied to the mirror descent algorithm. While there are differences between both settings, this significantly reduces the novelty of the analysis. Minor comments: 3. In Theorem 1, D is severely overloaded. It means both the arbitrary constant D as well as the Bregman distance, making for example the definition of C_D confusing. I encourage the authors to choose a different name for D. 4. In Eq. (4.7), it's unclear if the sum of \gamma_i^2 is in the numerator or denominator. Please use parenthesis to make this clear. 5. Below Eq. (5.2), is rho = theta? Otherwise I don't see any \rho in the formula. # Post-rebuttal I have read the author's rebuttal and the other reviewer's comments. The author have clarified the relationship with paper ID #4247 which was my main criticism. I have increased my score accordingly.

Reviewer 3



This paper proposes a regularity condition together with an adaptive mirror-prox algorithm aiming to solve VI problem with possibly singular operators. They recover the optimal rate of $O(1/T)$ in the deterministic case by replacing Lipchitz continuity with the proposed one, both for MP and AMP. Also, they prove the $O(1/\sqrt(T))$ convergence rate of AMP in the stochastic case. The paper presents nice results but some are not surprising. Some issues in details: 1. The idea of Lipchitz-like condition is proposed by other works. Also, the Bregman parts are not novel. 2. The stochastic result $O(1/\sqrt(T))$ is basically mirror-prox under the proposed condition. The analysis of AMP covers the deterministic case only. So the statement Line 13-15 is kind of misleading. At the same time, the theory parts are not too strong.

[Author Response · NeurIPS 2019]

To begin, we would like to extend our thanks to the reviewers for their detailed and insightful remarks. We reply to the reviewers' major concerns below:

**Reviewer 1.**   We thank the reviewer for their positive evaluation and encouraging remarks!

1. Regarding the lack of numerical experiments: we would kindly refer the reviewer to Section D of the supplement where we perform a series of numerical experiments on the resource sharing problem discussed in the paper's body. In these experiments, we compare the convergence rate of Korpelevich's extra-gradient algorithm to mirror-prox with a suitable Bregman function, and our adaptive mirror-prox variant. We wanted to include these experiments in the main body of the paper, but we ran out of space – we would be happy to use the extra page to do so if the paper is accepted.

2. On the complexity of calculating mirror steps: indeed, in full generatlity, calculating mirror steps can be computationally expensive (just as Euclidean projections can be bottlenecks in many problems). As is common in the literature (see e.g., Nemirovski, 2004, Nesterov, 2007, Juditsky et al., 2011, and the many works citing these papers), we were tacitly assuming a "prox-friendly" setting where mirror steps can be computed efficiently. The example we presented is indeed prox-friendly because it only involves the calculation of a single Lagrange multiplier (which can be accomplished efficiently via a simple line search). We will gladly explain this in more detail in the paper.

3. Beyond monotonicity: excellent suggestion! Albeit outside the scope of this work, local convergence in non-monotone VIs is one of our future research agendas and we would be happy to discuss it in more detail.

**Reviewer 2.**   We thank the reviewer for their positive recommendation! We address their main concerns below:

1. On numerical experiments: please see Section D of our paper's supplement and our reply to Reviewer 1 above.

2. On adaptation via backtracking: this is a very interesting suggestion and we thank the reviewer for making it. Two particularly promising works in that direction are the 2006 JCAM paper of Y. He and the 2018 SIOPT paper of Malitsky and Pock. Carrying out a linesearch analysis along these lines lies beyond the scope of the current work, but we would be happy to discuss this promising agenda in detail in a revision.

3. On (4.2b): yes, this was supposed to refer to the dual of the local norm. Apologies for any confusion!

4. On the continuity condition of #4247: we believe there may have been a misunderstanding here. To put things on an equal footing, consider the problem of minimizing a given convex function $f$ (the common denominator of variational inequalities and online/stochastic optimization problems). The growth condition of #4247 extends the notion of **Lipschitz continuity of** $f$ to singular problems; by contrast, Bregman continuity (this paper) extends the notion of **Lipschitz continuity *of the gradient of* $f$\*** to singular problems.
   The optimization literature makes a clear distinction between these two settings (often referred to as "non-smooth" and "smooth" respectively), because they lead to very different algorithms and oracle complexity bounds – see e.g., Nesterov's (2004) classical textbook on the topic. In particular, relative to #4247: (*a*) the algorithms are different (mirror-prox vs. FTRL); (*b*) the rates obtained are different (mirror-prox achieves a $\mathcal{O}(1/T)$ convergence rate in deterministic minimization problems whereas, without further assumptions, FTRL only achieves $\mathcal{O}(1/\sqrt{T})$); and, of course, (*c*) as we stated above, the regularity conditions considered are clearly different, even in convex minimization problems.

5. On (5.2): yes, the $\rho$ below (5.2) should be $\theta$, thanks!

**Reviewer 3.**   We thank the reviewer for their positive evaluation and constructive feedback! Our replies follow below:

1. Regarding the "Lipschitz-like" condition of [6]: to clarify, this is an extension of Lipschitz gradient continuity for functions with singularities, and is designed to yield an appropriate descent lemma. Since there is no longer an objective function to minimize, descent lemmas are not applicable to variational inequality problems; as a result, we had to seek a different starting point altogether.

2. Regarding the statement in Lines 13-15: we see the reviewer's point, thanks! We did not mean for the sentences to be parsed in that order, we will of course change the phrasing accordingly.

3. Regarding the definition of the local norm in lines 170-171: by a "continuous assignment of a norm" we had in mind the definition of an absolutely homogeneous Finsler metric as in the classic textbook of Bao, Chern and Shen, 2000. Specifically, a local norm is a continuous nonnegative function $F: \mathcal{X} \to \mathcal{V} \to \mathbb{R}_+$ with the following properties for all $x \in \mathcal{X}$, $z_1, z_2 \in \mathcal{V}$: (*i*) $F(x, z_1 + z_2) \le F(x, z_1) + F(x, z_2)$; (*ii*) $F(x, \lambda z) = |\lambda| F(x, z)$; and (*iii*) $F(x, z) > 0$ whenever $z \in \mathcal{V} \setminus 0$. The "local norm" at $x$ would then be $\|z\|_x = F(x, z)$ in this language. We will be happy to provide this more detailed definition in the paper.

4. Regarding the numerical experiments in the supplement: AMP was run with $\theta = 0.9$. In practice, we saw very little difference for values of $\theta$ between $0.5$ and $0.99$.

[Meta-Review · NeurIPS 2019]

This paper presents an analysis of Mirror-Prox method under Bregman continuity (instead of Lipschitz continuity) and propose a method which does not require knowledge of the continuity parameter. This is a good addition to the literature on Mirror-Prox.